# Phytochemical Composition of the Fruits and Leaves of Raspberries (*Rubus idaeus* L.)—Conventional vs. Organic and Those Wild Grown

Marta Kotuła [1],*, Joanna Kapusta-Duch [1],*, Sylwester Smoleń [2] and Ivo Doskočil [3]

1 Department of Human Nutrition and Dietetics, Faculty of Food Technology, University of Agriculture in Krakow, Al. Mickiewicza 21, 31-120 Krakow, Poland
2 Unit of Plant Nutrition, Institute of Plant Biology and Biotechnology, Faculty of Biotechnology and Horticulture, University of Agriculture in Krakow, Al. Mickiewicza 21, 31-120 Krakow, Poland
3 Department of Microbiology, Nutrition and Dietetics, Faculty of Agrobiology, Food and Natural Resources, Czech University of Life Sciences Prague, Kamycka 129, 165 00 Praha-Suchdol, Czech Republic
* Correspondence: marta.kotula@urk.edu.pl (M.K.); joanna.kapusta-duch@urk.edu.pl (J.K.-D.)

**Abstract:** Nutrition is an influential determinant of the risk of present-day metabolic diseases. Raspberries (*Rubus idaeus* L.) are extraordinary berries with a high nutritional and bioactive component complex. They have a number of major essential minerals and trace elements as well as dietary fibre and other important constituents. This study aimed to analyse and compare raspberry fruits and leaves originated from organic versus conventional agricultural practices and wild grown for the contents of basic composition (water, crude fat, total protein, ash, digestible carbohydrates and dietary fibre) and selected minerals (calcium, iron, potassium, magnesium, sodium, phosphorus, sulphur, selenium, barium, lithium, and beryl) as well as selected antioxidant properties (total polyphenols, total carotenoids, anthocyanin content, and antioxidant activity by methods of ABTS and FRAP). This study was carried out regardless of climatic and agro-technical factors and was of a more consumer-oriented nature, in order to recognize the diversity of raspberry fruits and leaves from more or less monitored crops. The basic composition, mineral content and selected antioxidative properties of raspberry fruits and leaves are fundamentally different. Raspberry fruits have a lower content of protein and ash, and higher levels of dietary fibre and carbohydrates in comparison to fruits. The biggest difference is the amount of protein, whose content in leaves is two to three times higher versus fruits. Raspberry leaves have been found to have a higher mineral content than raspberry fruits and were characterized by up to five times the amount of total polyphenols, with respect to raspberry fruit, regardless of source. The content of total carotenoids was found in some cases to be 100 times higher in raspberry leaves, in comparison to fruit, regardless of origin. It has not been definitely identified, both for raspberry fruits and leaves, which method of growing is the most advantageous in terms of levels of basic nutrients, selected minerals and antioxidant properties.

**Keywords:** organic cultivation; wild berries; conventional farming; major essential minerals; trace elements; antioxidant properties

## 1. Introduction

Berries, thanks to their distinctive color, specific taste and aroma, high content of vitamins and minerals, as well as their wide use in the food process industry, play an important role among other fruits. Moreover, they can grow in extreme conditions, in which a majority of other fruit species do not grow. They are widely distributed. A rich variety of species grows in the North America, particularly in the USA and Canada [1,2]. *Rubus* L. (*Rosaceae*) berries, due to their nutritional and bioactive value, arouse interest all over the world. Red raspberry (*Rubus idaeus* L.), an herbaceous plant in the *Rosaceae* family, is a bush described as a nanofanerophyte. Raspberries are prized by both nutritionists and

herbalists, due to their high content of pro-health substances [3]. A chemical constitution of raspberries leaves, which are a by-product of raspberry processing, has not been studied as thoroughly as its fruit. In particular, the healing properties of raspberry leaves, known since antiquity, are prescribed for the treatment of a wide variety of diseases, for example, by including them in herbal preparations used for the uterus relaxation during childbirth. Other applications of raspberry leaves include their use as an additive to drinks, nutritional supplements, preparations of functional herbal teas, teas, and chocolate, in order to enhance their nutritive and flavour-forming properties [4,5].

It has been revealed that *Rubus* berries are an increasingly important source of bioactive substances due to their anti-oxidant, anti-inflammatory, chemopreventative and antibacterial properties, their positive effects on blood lipids and above-mentioned atherosclerosis, as well as their advantageous composition. Therefore, thanks to their biological effect, they can potentially be applied as health-promoting [6].

The development of an increasing number of diseases has been attributed to the action of oxygen free radicals. Some of these diseases are caused by a shift in the cell's oxidoreductive balance toward excessive free radical production by the mitochondria or as a result of uncontrolled inflammation (e.g., diabetes, atherosclerosis, or rheumatoid arthritis). Others, such as cancer, are due to disruptions in intracellular signal transduction or gene expression. Still others are due to impaired antioxidant mechanisms, particularly a reduction in GSH (glutathione) or impaired SOD (superoxide dismutase) activity. Free radicals are responsible for complications in transplantation, as well as heart attacks and strokes; they also lead to the development of Parkinson's and Alzheimer's diseases, some prenatal complications, lung and intestinal diseases, cataracts, male infertility, and poisoning [7,8].

There are non-enzymatic and enzymatic mechanisms in the body that protect against the excess and harmful effects of free radicals. Non-enzymatic mechanisms are based on chemical reactions mainly involving vitamin E in cell membranes, vitamin C in the cytoplasm, and phenolic compounds. Scientists emphasize the biological importance of polyphenolic compounds as effective antioxidants. These are secondary metabolites widespread in the plant world (mostly in the form of glycosides or esters) that are not synthesized in animal organisms. Polyphenols can act in several ways: as reducing agents, as free radical blocking compounds, chelating metal ions that catalyze oxidation reactions (thus preventing reactions caused by a single active oxygen atom), and inhibiting the activity of oxidative enzymes such as lipoxygenases. Polyphenols have an ideal chemical structure for capturing free radicals and for forming chelates with metals, making them much more effective in vitro antioxidants than antioxidant vitamins [9,10]. Carotenoids exhibit a broad spectrum of antioxidant activity, and they are classified as both preventive and interventional antioxidants. They are divided into two subgroups: carotenes and xanthophylls. Metals, which are activators or components of important enzymes, i.e., selenium—glutathione peroxidase, iron—catalase, copper, zinc, and manganese—superoxide dismutase, play an important role in enzymatic mechanisms that protect against the excess and harmful effects of free radicals. Therefore, the best way to prevent diseases resulting from free radicals is to consume products rich in natural antioxidants [8]. Raspberries are among the fruits that are rich in such bioactive substances.

The wild edible fruit species are an important component of biodiversity. Raspberries also often grow in the wild, especially in woodlands, glades, grasslands, and on mountainsides.

This work discusses and compares the basic composition, as well as the presence of selected minerals, in wild raspberry, in order to be able to compare them with raspberry from controlled crops (i.e., ecological and conventional).

The contribution of ecological farming to environmental protection is a total prohibition on the use of synthetic chemicals in agriculture and the promotion of natural growing methods. The differences found in the amount of bioactive components in crops result mainly from the agricultural practices used in various agri-systems [11].

Therefore, this study aimed to examine fruits and leaves of raspberries derived from conventional, organic, and wild grown sources for the contents of water, crude fat, total protein, ash, digestible carbohydrates and dietary fibre, and selected major essential minerals as well as trace elements (calcium, iron, potassium, magnesium, sodium, phosphorus, sulphur, selenium, barium, lithium, and beryl). With this goal in mind, it has been hypothesized that the basic composition and selected minerals content in the fruit and leaves of wild-grown raspberries, as well as those from ecological farming and randomly obtained from local stores (conventional), varied significantly, independent of both climatic and agrotechnological factors.

The existing literature shows that data on the content and identification of heavy metals in the fruits and leaves of raspberries (*Rubus idaeus* L.) have already been published, but there are no studies of some elements (e.g., barium, lithium, selenium, and beryl) and they rarely cover two years in cultivation. The secondary significant point that has not yet been reported in the published literature is the comparison of these two different types of cultivation, that is, conventional and organic, also in relation to wild-growing ones under similar climatic conditions, which are a possible alternative for the consumer, but are not yet fully understood.

## 2. Materials and Methods

### 2.1. Red Raspberry Fruits and Leaves

The test materials were raspberry (*Rubus idaeus* L., variety Polana) fruits and leaves from different agricultural practices (conventional vs. organic), as well as wild grown (unknown variety). The material came from a 2-year collection (August 2020 and August 2021). The conventional raspberries were obtained from the farm in the village Jadowniki (Malopolska Voivodeship 49.9588784, 20.643984269828643), while the ecological raspberries originated from a certified cultivation in the village Bębło (near Krakow, Malopolska Voivodeship 50.18166317193472, 19.78522432981779). Both farms were located in similar climatic and soil conditions, so as to minimize the number of factors that may affect the quality of the examined raspberry varieties. Another factor was the choice of variety i.e., *Rubus idaeus* L., *Polana* cv. The wild raspberry was harvested in the woods near the countryside of Siedlec (close to the town Częstochowa) in the Siesian Voivodeship 50.68940063565632, 19.367067526363734. Therefore, the examined raspberries were growing in the same conditions as the raspberries that every consumer of organic and conventional fruit can buy.

The test material was collected at the phase of harvest maturity. Pre-processing of directly harvested raspberry fruits and leaves included selection and cleaning. Then these samples were frozen at −22 °C in a Liebherr GTS 3612 chamber freezer (Germany). The next step was freeze-drying using a Christ Alpha 1–4 freeze-dryer and milled in a Tecator Foss grinder (Hillerød, Sweden).

### 2.2. Analysis of Basic Composition and Selected Minerals

Analytical Methods

Dry matter content was estimated as a loss of mass on drying at 105 °C at normal conditions of pressure in accordance with Standard PN-EN-12145:2001 [12].

Both fresh and freeze-dried fruits and leaves were also examined for contents of total protein (using Tecator Kjeltec 2200 system—Tecator, Sweden); crude fat (using Tecator Soxtec Avanti 2050 -Tecator, Sweden); ash by dry mineralisation at a temperature of 525 °C in a muffle furnace [13], and dietary fibre by an enzymatic-gravimetric procedure (using Tecator Fibretec System E—Tecator, Sweden).

The crude protein amount was found in accordance with International Standard PN-EN ISO 20483:2014-02 [14]. During this process, the sample was submitted to aqueous mineralization in condensed sulfuric acid of a certain analytical grade (Chempur, Poland), then alkalization and distillation of the liberated ammonia. The nitrogen concentration was then determined. Total protein levels were evaluated by calculating the nitrogen content by

multiplying the nitrogen-to-protein conversion factor of 6.25 (corresponding to 0.16 g of nitrogen per gram of protein).

Crude fat content was estimated by extracting fat from dehydrated material using a petroleum ether of analytical grade (Chempur, Poland) according to the International Standard PN-EN ISO 11085:2015-10 [15]. The solvent was then vaporized and the residue dried at 105 °C. Lastly, the weight of the recovered fat was used to calculate the crude fat concentration.

Dietary fibre was determined by means of enzymatic-gravimetric method upon the principles of AOAC International Official Method 985.29 [16]. Freeze-dried samples of raspberry fruits and leaves were gelatinized by a temperature-stable α-amylase. Then, to remove starch and protein, protease and amyloglucosidase were added. Afterwards, ethyl alcohol was added to form precipitate, which was next filtered and cleaned in ethyl alcohol and acetone, then dried and finally weighed. One portion was used for protein determination, while the other was incinerated.

Digestible carbohydrate content was counted with the equation: digestible carbohydrate = 100 − (crude fat + protein + ash + dietary fibre).

Micronutrients and trace elements were measured in 0.5 g air dry samples, which were placed into 55 mL TFM dishes and then mineralized in 10 mL of $HNO_3$ (65% super pure, Merck no. 100443.2500) with the Mars 5 Xpress (CEM, USA) microwave mineralization system. The digestion procedure was as described below. 15 min, to achieve the temperature of 200 °C; and 20 min, to keep this temperature. After that, samples were chilled and quantitatively moved to 25 mL volumetric flasks with redistilled water. The amounts of magnesium (Mg), calcium (Ca), sulphur (S), potassium (K), phosphorus (P), sodium (Na) and iron (Fe) were quantified using an optical emission spectrometer with high-dispersion inductively coupled plasma (ICP-OES by Prodigy Teledyne Leeman Labs, USA) [17]. Since selenium (Se), lithium (Li), barium (Ba) and beryl (Be) are all of lower densities, their analysis was performed using inductively coupled plasma mass spectrometry with a triple quadruple spectrometer (iCAP TQ ICP-MS by ThermoFisher Scientific, Bremen, Germany).

### 2.3. Analysis of Antioxidant Activity

#### 2.3.1. Preparation of Methanol Extracts

Methanol extracts (5 g of raw fruits and 2 g of raw leaves in 80 mL of a 70% methanol solution) were obtained from a mean representative sample by shaking the fresh plant material for 2 h at the room temperature in a water bath shaker type 357, Elpan (Lubawa, Poland). They were then filtered using filter paper and finally stored at −22 °C [18].

#### 2.3.2. Total Polyphenolic Compounds Concentration

The above-described methanolic extracts were used to establish the total polyphenolic compounds content, using the Folin-Ciocalteau reagent (Sigma-Aldrich, St. Louis, MO, USA) [19]. The methanol extracts were diluted 1:20 with distilled water. The content of total phenols in the 5 mL diluted extracts was measured after 20 min by means of a spectrophotometric method at 760 nm with a 0.5 mL of Folin-Ciocalteu reagent and 0.25 mL of 25% sodium carbonate using a RayLeigh UV-1800 spectrophotometer (China). The results were expressed as milligrams of chlorogenic and gallic acid equivalents per 100 g of dry weight, followed by CGA and GAE based on the standard curves for the above-mentioned acids (mg CGA and GAE/100 g d.m.).

#### 2.3.3. Anthocyanins Concentration

The concentration of anthocyanins was determined using the Fuleki and Francis method, which involves measuring the difference in absorbance of two buffer solutions at pH 1.0 and pH 4.5 at wavelengths of 524 and 700 nm using RayLeigh UV-1800 spectrophotometer (China) according to [20]. The starting material was methanol extracts from which anthocyanins were extracted with a combination of methanol and HCl (95:5 0.1 N HCl). The results are reported per cyanidin-3-glucoside.

### 2.3.4. Determination of Antioxidant Activity

Identical methanolic extracts were used to determine the antioxidant activity based on ABTS$^{\bullet+}$ free radical (2,2'-azinobis-(3-ethylbenzothiazoline-6-sulfonic acid) and FRAP (ferric-reducing antioxidant power).

### 2.3.5. ABTS Method

The ability to scavenge the ABTS$^{\bullet+}$ free radical is performed by a colorimetric assay of the content of the ABTS$^{\bullet+}$ free radical solution, which had not been reduced by the antioxidant present in the products examined [21]. This in vitro assay involves the generation of a relatively stable free radical that loses color after scavenging electrons from lipophilic and hydrophilic antioxidants in a sample. ABTS (2,2'-azinobis-(3-ethylbenzothiazoline-6-sulfonic acid), potassium persulfate and Trolox were purchased from Sigma-Aldrich (Sigma-Aldrich, St. Louis, MO, USA). The color change, monitored by the change in absorbance at 734 nm via a spectrophotometer (UV-1800, Rayleigh, Beijing, China) after a specified time and temperature (6 min at 30 °C), is proportional to the antioxidant's concentration.

### 2.3.6. FRAP Method

Ferric reducing antioxidant power (FRAP) assay uses antioxidants as reductants in a redox-linked colorimetric reaction, wherein $Fe^{3+}$ is reduced to $Fe^{2+}$. Ferric ($Fe^{3+}$) to ferrous ($Fe^{2+}$) ion reduction at low pH causes formation of a colored ferrous-probe complex from a colorless ferric-probe complex.

The FRAP analysis was performed in accordance with [22]. TPTZ (2,4,6-Tris(2-pyridyl)-s-triazine), HCl, iron chloride and reagents for acetic buffer (sodium acetate and glacial acetic acid) were purchased from Sigma-Aldrich (Sigma-Aldrich, St. Louis, MO, USA). Samples were combined with the reaction mixture previously mixed in the ratio of 10:1:1 (acetic buffer, TPTZ solution in 0.40 M HCl, iron chloride solution in distilled water). They were then measured after 10 min at 593 nm with a RayLeigh UV-1800 spectrophotometer (Beijing, China).

The values obtained for each sample in ABTS and FRAP method, after their comparison with the concentration–response curve of the standard Trolox solution, were expressed as µmol Trolox equivalents per gram of dry weight (TEAC) (µmol Trolox/g dry weight).

### 2.3.7. Total Carotenoids Compounds Concentration

Total carotenoids content was estimated by extracting carotenoids from samples using a combination of acetone and hexane (4:6 *v/v*) (Chempur, Piekary Śląskie, Poland) following the Polish Standard with slight adaptations [23]. Samples with either 0.5 g (fruits) or 0.15 g (leaves) of dried raspberries were weighted in a porcelain mortar. The pigment was then extracted using a mixture of acetone and hexane with approximately 0.5 g of sand. The extracts were transferred to a cylinder and, next, the volume of the extract was measured. The absorbance was tested after 30 min at 450 nm via a spectrophotometer (UV-1800, Rayleigh, Beijing, China). The values of the results were evaluated using a β-carotene standard curve (Sigma-Aldrich, St. Louis, MO, USA).

### 2.4. Statistical Analysis

All measurements were carried out in at least three replications and results were presented as means ± standard deviations (SD). A two-way analysis of variance and Duncan's test at $\alpha \leq 0.05$ were made to compare significant differences in mean values. Two factors were cultivation and harvest time. To perform all evaluations, Statistica software v. 13.1 PL (Dell Inc., Tulsa, OK, USA) was applied.

## 3. Results and Discussion

### 3.1. Basic Composition of Raspberry Fruit

The basic composition of raspberry (*Rubus idaeus* L.) fruits from various kinds of cultivation is shown in Table 1. Due to the different size of the fruit, and thus the water

content, the results were converted to dry matter. The fruits of wild-growing raspberry have a smaller size and a lower water content, which agree with the findings of Dubravka [24]. In turn, dry matter content of conventional and organic raspberries does not differ significantly.

**Table 1.** Basic composition in fruits of raspberries from conventional and organic practices as well as those wild grown (g 100 g$^{-1}$ dry weight—DW).

| Basic Composition | Cultivation | Conventional | | Organic | | Wild Crop | |
|---|---|---|---|---|---|---|---|
| | Year | 2020 | 2021 | 2020 | 2021 | 2020 | 2021 |
| Water | | 83.52 ± 0.06 [c] | 87.04 ± 0.25 [b] | 84.79 ± 0.13 [d] | 86.06 ± 0.26 [b] | 81.06 ± 0.46 [a] | 81.17 ± 0.90 [a] |
| Total protein | | 7.03 ± 0.10 [a,b] | 6.78 ± 0.13 [a] | 9.01 ± 0.21 [c] | 7.17 ± 0.02 [b] | 9.16 ± 0.09 [c] | 6.95 ± 0.09 [a,b] |
| Crude fat | | 3.98 ± 0.24 [a] | 3.21 ± 0.10 [c] | 5.88 ± 0.35 [b] | 4.73 ± 0.13 [d] | 5.55 ± 0.30 [b] | 4.14 ± 0.22 [a] |
| Ash | | 2.55 ± 0.01 [b] | 3.48 ± 0.11 [d] | 2.93 ± 0.10 [a] | 2.97 ± 0.05 [a] | 2.90 ± 0.13 [a] | 3.27 ± 0.03 [c] |
| Dietary fibre | | 33.45 ± 0.63 [a] | 38.28 ± 0.11 [b,c] | 36.84 ± 0.81 [b] | 34.16 ± 0.63 [a] | 49.31 ± 0.80 [d] | 38.66 ± 0.19 [c] |
| Digestible carbohydrates | | 86.44 ± 0.13 [a] | 86.53 ± 0.14 [a] | 82.18 ± 0.66 [b] | 85.13 ± 0.10 [c] | 82.37 ± 0.52 [b] | 85.64 ± 0.09 [a,c] |

Results (mean ± SD) obtained from the analysis of three individual samples (n ≥ 3). Values within lines with different letters are significantly different (Duncan test $p \leq 0.05$).

There are several hundred types of raspberry known in the world, and their number is constantly growing. Several dozen are cultivated on a larger scale. In Poland, only seedlings of the varieties entered in the register may be sold, because it is an economically important species. Nursery material is produced under the control of the State Seed Inspection and the State Plant Protection Service. The wild grown are not under control. We do not know exactly the varieties of wild growing *Rubus idaeus* species. This makes it a bit more difficult to discuss with authors who have conducted research on known varieties.

The amounts of total protein and crude fat were significantly higher ($p \leq 0.05$) in fruits of raspberries from organic farming and those wild-grown collected in 2020, compared to those from conventional agriculture practice. As for these constituents, the relationship found for the raspberries harvested in 2021 was similar and generally significant. The only difference was the comparable content of total protein in conventional and wild grown raspberries. The highest significant ash content was noted not only in conventional and wild-growing raspberries from the harvest 2021, compared to organic ones, but also to conventional and wild-growing raspberries harvested in 2020, for which the results were similar. The highest significant dietary fibre amounts were found in both wild-growing raspberries harvested in 2020 and those collected in 2021 compared to other examined raspberries. In 2020, its content in wild-growing raspberry fruits was almost 50 g/100 g DW. Digestible carbohydrates were determined in the highest, statistically significant amounts ($p \leq 0.05$), in conventional raspberries from 2020 and 2021 as well wild-growing ones from 2021, compared to remaining fruits, for which the results were similar ($p > 0.05$).

As for protein, its higher levels were noted by Jeong, et al. [25] in commercial raspberries from Korea, in comparison to the amounts obtained in this work. Similar results to those achieved in this paper of protein content were reported by Vara, et al. [26] and de Souza, et al. [27], while lower values were determined by other researchers [28]. The fat content of raspberries determined by other authors is varied, which is due to the amount of seeds contained in the fruit [29]. Jeong, et al. [25] found that the fat content in raspberries coming from Korea was 7.37%, while significantly a lower value, of 0.78 g per 100 g DW (dry weight), was obtained by Vara [26]. According to other authors, the determined fat content was similar or lower [27]. In the examined samples, the ash content ranged from 2.55% to 3.48%. The ash content given by Vala, et al. [26] was much the same as that of conventional and wild-growing raspberries in the second year of cultivation, and was 3.9% DW. On the other hand, the results shown by other authors were significantly higher [25]. Similar results for this nutrient were reported by Akimov, et al. [28] although other authors stated lower ash contents [27]. The examined raspberry fruits had a considerable amount of fibre, from 33.45 to 38.28 g in conventional and ecological crops and up to

49.31 g per 100 g DW in wild-growing raspberry. However, the result reported by Jeong, et al. [25] was significantly lower than that obtained in this study. According to Akimov, et al. [28], the total fibre content in the raspberry cultivar Polana was 31.25% DW, of which 68.6% was insoluble fibre fraction. Of berries such as blackberry, blueberry or strawberry, raspberry fruits have the highest amounts of total fibre and its insoluble fraction, which is particularly important for nutritional reasons [30]. A high fibre content (59.5%) was also stated by Brodowska [31], however, the examined material was raspberry pomace. In other studies, similar results for fibre content were reported [27]. Considering carbohydrate content, the values obtained in this study for the examined raspberries were similar; those from conventional farming were slightly higher. Similar levels of this macronutrient were noted by other researchers [27]. Higher carbohydrate contents were determined in the fruit of the raspberry cultivar Kweli [26].

### 3.2. Selected Minerals Content in Raspberries Fruits

Mineral content of raspberry fruits from various kinds of cultivation is shown in Table 2.

**Table 2.** Selected minerals in fruits of raspberries from conventional and organic practices as well as those wild grown (mg 100 g$^{-1}$ DW).

| Mineral Content | Cultivation | Conventional | | Organic | | Wild Crop | |
|---|---|---|---|---|---|---|---|
| | Year | 2020 | 2021 | 2020 | 2021 | 2020 | 2021 |
| Calcium (Ca) | | 116.00 ± 25.80 [c] | 157.36 ± 11.64 [a] | 193.59 ± 7.76 [b] | 178.91 ± 3.87 [a,b] | 175.17 ± 8.71 [a,b] | 225.12 ± 9.65 [d] |
| Iron (Fe) | | 3.60 ± 0.01 [a] | 3.48 ± 0.36 [a] | 4.85 ± 0.34 [b] | 4.39 ± 0.38 [b] | 3.71 ± 0.11 [a] | 7.65 ± 0.37 [c] |
| Potassium (K) | | 926.59 ± 187.84 [a] | 1299.45 ± 92.61 [b] | 998.38 ± 210.69 [a] | 1311.97 ± 69.18 [b] | 789.24 ± 35.18 [a] | 895.80 ± 38.84 [a] |
| Magnesium (Mg) | | 137.59 ± 20.74 [a] | 159.11 ± 4.08 [c] | 138.51 ± 5.35 [a] | 156.21 ± 6.83 [c] | 119.88 ± 1.44 [b] | 133.15 ± 1.74 [a,b] |
| Sodium (Na) | | 3.04 ± 0.47 [c] | 0.02 ± 0.00 [b] | 1.08 ± 0.03 [a] | 0.06 ± 0.01 [b] | 1.31 ± 0.29 [a] | 0.99 ± 0.06 [a] |
| Phosphorus (P) | | 152.14 ± 19.56 [a] | 268.15 ± 6.58 [d] | 156.53 ± 3.35 [a] | 248.02 ± 9.61 [c] | 163.61 ± 2.55 [a] | 206.04 ± 5.98 [b] |
| Sulphur (S) | | 71.88 ± 9.64 [c] | 81.41 ± 1.57 [a,b] | 75.97 ± 3.58 [a,c] | 97.65 ± 2.94 [d] | 83.59 ± 3.33 [a,b] | 90.01 ± 2.76 [b,d] |
| Selenium (Se) | | nd | 0.0008 ± 0.000 [a] | nd | 0.0004 ± 0.000 [a] | nd | 0.0006 ± 0.000 [a] |
| Barium (Ba) | | 0.26 ± 0.01 [c] | 0.16 ± 0.01 [a] | 0.33 ± 0.02 [d] | 0.21 ± 0.01 [b] | 1.15 ± 0.03 [e] | 1.65 ± 0.02 [f] |
| Lithium (Li) | | 0.0028 ± 0.003 [b] | 0.0001 ± 0.000 [a] | 0.0010 ± 0.000 [a,b] | 0.0003 ± 0.000 [a,b] | 0.0003 ± 0.000 [a,b] | 0.0014 ± 0.000 [a,b] |
| Beryl (Be) | | 0.0002 ± 0.000 [a,b] | 0.0001 ± 0.000 [b] | 0.0003 ± 0.000 [a,c] | 0.0002 ± 0.000 [a,b] | 0.0003 ± 0.000 [a,c] | 0.0004 ± 0.000 [c] |

Results (mean ± SD) obtained from the analysis of three individual samples (n ≥ 3). Values within lines with different letters are significantly different (Duncan test $p \leq 0.05$); nd—not detected.

In comparison with all examined fruit, calcium content was the highest ($p \leq 0.05$) in wild-growing raspberries harvested in 2021 and organic raspberries from the 2020 harvest. The lowest statistically significant ($p \leq 0.05$) content was found in conventional raspberries harvested in 2020 compared to the remaining ones. Wild-growing raspberries harvested in 2021 had the highest statistically significant ($p \leq 0.05$) iron content, compared to other samples. In both harvest years, organic raspberries also contained significantly more iron ($p \leq 0.05$) than conventional and wild raspberries harvested in 2020. The contents of potassium, magnesium, and phosphorus were the highest ($p \leq 0.05$) in conventional and organic raspberries harvested in 2021, compared to other samples, for which these values were similar ($p > 0.05$). In both conventional and organic raspberries from 2020 and wild-growing raspberries from both harvests, sodium contents were significantly higher ($p \leq 0.05$) than in conventional and organic raspberries from 2021, containing similar ($p > 0.05$) amounts. The highest statistically significant sulphur content ($p \leq 0.05$) was found in organic raspberries harvested in 2021, while the lowest was in conventional raspberries from the 2020 harvest, compared to the remaining raspberries. Selenium content in raspberries was determined only in 2021; the results obtained for all compared crops were similar ($p > 0.05$). As for barium, its content in wild-growing raspberries was significantly higher ($p \leq 0.05$) than in other samples, regardless of the harvest year. The

lowest ($p \leq 0.05$) content of this element was determined in conventional raspberries harvested in 2021. Lithium was statistically present at the highest concentration ($p \leq 0.05$) in conventional raspberries from the 2020 harvest, and the lowest significant content in conventional raspberry as well, compared to the other samples analysed, where the results were comparable ($p > 0.05$). The highest statistically significant ($p \leq 0.05$) beryllium concentration, compared to other samples, was found in wild-growing raspberries, while the lowest was in the conventional raspberries harvested in 2021.

As for calcium, numerous authors have identified its higher levels [25,28,32]. Calcium levels given by De Souza, et al. [27], amounting to 100 mg 100 g$^{-1}$ DW, were significantly lower than those determined in this study. The iron content, reported by Akimov, et al. [28], corresponded to our results obtained for conventional and wild-growing raspberries from the first year of the experiment (3.84 mg 100 g$^{-1}$ DW). Miliković, et al. [33] in his study on the organic and conventional raspberry cultivar Willamette, found lower iron content compared to our results. However, its amount in the organic fruits from the second year of cultivation increased significantly from 0.95 to 3.11 mg 100 g$^{-1}$ DW. Pavlović, et al. [32] proved that different raspberry cultivars have different iron contents. This can be explained by the iron content in the environment, since raspberries accumulate their large amounts of iron from the environment [34]. Raspberry fruits from Brazil had a much higher iron content, amounting to 9.3 mg 100 g$^{-1}$ DW [27]. A similar level of potassium was shown by other authors [25,28,32], while raspberries from Brazil contained lower amounts of this element, amounting to 603.2 mg 100 g$^{-1}$ DW [27]. The magnesium contents determined by Akimov, et al. [28] and Jeong, et al. [25] were 93.75 mg 100 g$^{-1}$ and 91.57 mg 100 g$^{-1}$ respectively, and were slightly lower than ours results. Some authors found higher Mg amounts [32], while similar contents were also stated [27]. The sodium content varied in the second harvest year of conventional and organic raspberries. Moreover, other studies showed a much higher sodium content than our findings [25,28,35]. The sodium content in the cultivars Meeker and Willamette, reported by Pavlović, et al. [32] corresponded to that found in our raspberry fruits in the first year of harvest. Furthermore, the cultivar Willamette from Zlatibor had a low sodium content, similar to the results obtained for raspberries from the second year of cultivation [32]. According to Jeong, et al. [25] phosphorus content in raspberry fruit was markedly higher, of 1297.09 mg 100 g$^{-1}$ DW; although a lower phosphorus content was also determined [27]. Castilho Maro, et al. [36] examined the raspberry cultivar *Polana* grown in Brazil and showed that raspberries had a significantly lower content of sulphur than our results.

Table 3 presents basic composition and minerals contained in raspberry fruits, based on the databases of the U.S. Department of Agriculture (USDA) [37] and the Polish database of Kuchanowicz, et al. [38].

When comparing the values obtained with the FoodData Central database [37] and the Polish database of Kunachowicz, et al. [38], the largest difference is in fat content, which was three times to twice as low as in this study. According to Kunachowicz, et al. [38], the content of dietary fibre in raspberry fruit is 47.18 g 100 g$^{-1}$. In this research, only the fruits of wild grown raspberries, collected in 2020, had a similar dietary fibre content; the remaining results were much lower—from 33.45 to 38.66 g 100 g$^{-1}$. The remaining macronutrients did not differ significantly or were within the ranges of the databases mentioned. The contents of individual minerals in the investigated databases differ from each other. Raspberry fruits from the FoodData Central database by USDA [37] have a lower content of calcium, iron, potassium and phosphorus and a higher sodium content. In the examined raspberry fruits, the levels of calcium, iron, potassium and sodium were lower when compared to those investigated by Kunachowicz, et al. [38]. The mineral content of the raspberries analysed was closer to that given by the FoodData Central database by USDA [37], apart from the sodium content, which was higher in both databases, compared to our results. The is no data on the content of selenium, barium, lithium and beryllium in any database.

**Table 3.** Composition of raspberry fruit based on data from FoodData Central—US Department of Agriculture (USDA) [37] and Kunachowicz, et al. [38] (basic composition expressed as g 100 g$^{-1}$ DW, minerals in mg 100 g$^{-1}$ DW).

|  | USDA [37] | Kunachowicz, et al. [38] |
|---|---|---|
| Moisture | 85.6 | 85.8 |
| Protein | 7.01 | 9.15 |
| Fat | 1.32 | 2.11 |
| Ash | 2.43 | 4.23 |
| Fibre | - | 47.18 |
| Carbohydrates | 89.58 | 84.51 |
| Calcium | 111.11 | 246.48 |
| Iron | 3.13 | 5.63 |
| Potassium | 1083.33 | 1429.58 |
| Magnesium | 133.33 | 140.85 |
| Sodium | 17.36 | 14.08 |
| Phosphorus | 187.50 | 232.39 |

*3.3. Bioactive Compounds in Raspberries Fruits*

The content of total polyphenols, carotenoids, anthocyanins and antioxidant activity of raspberry fruits from various kinds of cultivation in different crops is shown in Table 4.

The highest content of total polyphenols (in conversion to chlorogenic) was observed by organic and wild grown raspberry fruits from 2020, compared to the other raspberry fruits that were analyzed. The lowest statistically significant amount of this compound was found by conventional raspberry fruits from 2020, according to the other raspberry fruits subjected to analysis. The highest content of anthocyanins was determined in organic and conventional raspberry fruits from 2021 versus the other raspberry fruits that have been analyzed. In contrast, the lowest statistically significant content of the discussed compounds was found in conventional raspberry fruits collected in 2020, when compared to the other analyzed sources. Antioxidant activity, measured by the ability to quench the cation radical ABTS$^{\bullet+}$, was greatest in conventional raspberry fruits from 2021 versus the other examined raspberry fruits. The lowest ABTS$^{\bullet+}$ free radical quenching capacity was found in raspberry fruits harvested in 2020, regardless of the source of origin. When antioxidant activity was tested by the FRAP method, the results were similar, though not the same, as the ABTS determination. The highest antioxidant activity according to this method was found in conventional and wild raspberry fruit from 2021, and the lowest in conventional raspberry fruit than in the other studied raspberry fruits. The greatest statistically significant content of total carotenoids was characterized by organic raspberries harvested in 2020 and 2021, in relation to the other raspberry fruits examined. The lowest content of these compounds, relative to the other raspberry fruits, was found in wild raspberry fruits.

**Table 4.** Total polyphenol, carotenoids, and anthocyanins content, as well as antioxidant activity of the fruits of three raspberries cultivations at different harvest (2020 and 2021).

| Bioactive Compounds | Cultivation | Conventional | | Organic | | Wild Crop | |
|---|---|---|---|---|---|---|---|
| | Year | 2020 | 2021 | 2020 | 2021 | 2020 | 2021 |
| Total polyphenols [mg CGA 100 g$^{-1}$ DW] | | 1290.87 ± 116.41 [b] | 1494.54 ± 27.12 [a,b,c] | 1673.49 ± 268.49 [a] | 1346.84 ± 18.85 [b,c] | 1630.69 ± 109.23 [a] | 1544.75 ± 26.24 [a,c] |
| Total polyphenols [mg GAE 100 g$^{-1}$ DW] | | 1519.62 ± 121.71 [c] | 1782.47 ± 28.25 [a,b] | 1933.09 ± 278.39 [b] | 1610.25 ± 22.12 [a,c] | 1850.57 ± 113.93 [a,b] | 1759.81 ± 27.38 [a,b] |
| Anthocyanins [mg 100$^{-1}$ g DW] | | 158.11 ± 2.90 [c] | 308.25 ± 0.73 [d] | 254.80 ± 42.55 [b] | 393.42 ± 2.78 [e] | 203.02 ± 2.05 [a] | 218.91 ± 4.43 [ab] |
| ABTS $^{+}$ [µmol Trolox g$^{-1}$ DW] | | 116.39 ± 7.60 [a] | 214.37 ± 2.18 [d] | 118.91 ± 0.73 [a] | 191.40 ± 2.82 [c] | 120.04 ± 1.09 [a] | 181.47 ± 0.48 [b] |
| FRAP [µmol Trolox g$^{-1}$ DW] | | 228.88 ± 21.91 [c] | 390.51 ± 1.19 [b] | 301.51 ± 44.34 [a] | 338.14 ± 9.82 [a,b] | 321.34 ± 22.83 [a] | 395.45 ± 15.44 [b] |
| Total carotenoids [mg 100 g$^{-1}$ DW] | | 1.78 ± 0.00 [e] | 1.59 ± 0.02 [c] | 1.96 ± 0.03 [a] | 1.91 ± 0.00 [a] | 1.33 ± 0.03 [b] | 1.69 ± 0.02 [d] |

Results (mean ± SD) obtained from the analysis of three individual samples (n ≥ 3). Values within lines with different letters are significantly different (Duncan test $p \leq 0.05$).

Stamenković, et al. [39] showed that the fruit of *Rubus idaeus* L. raspberry variety *Polana* had a polyphenol content of 1635.60 mg per 100 g of dry matter, which corresponds to the results we obtained. Russian authors examining the fruits of the *Polana* red raspberry have determined the total content of polyphenols (converted into gallic acid) similar to the results of raspberries from conventional cultivation (both cultivation years) and organic cultivation from 2021 [40]. The similar climatic and agrotechnological factors may indicate similarities in the obtained results. Other sources, on the other hand, report that the content of these compounds is $662.3 \pm 113$ mg in 100 g of fresh fruit weight (converted to gallic acid) [41]. Lower results were obtained by [42] who tested two raspberry cultivars *Rubus idaeus* L. *Indian Summer* and *Skeena* obtained in Japan, which had successively 4 to 6 lower milligrams of polyphenols content in relation to gallic acid, respectively.

Further sources state that, on the basis of a spectrophotometric method using the Folin-Ciocalteau reagent as a chromogen, the total concentration of polyphenolic compounds in *Rubus idaeus* L. was shown to be $140.6 \pm 0.9$ mg/100 g for early varieties, while $214.4 \pm 0.8$ mg/100 g for late varieties. The authors of the cited studies confirmed that not only the genotype of the fruit influences the content of polyphenols, but also documented that the period of fruit harvesting is of great importance. It turned out that fruits from later harvests contained significantly more of these compounds [43]. In the present study, the content of total polyphenols in wild raspberry is the highest, as mentioned earlier; this may be due to the highest dry matter content in these plants. Probably, the raspberries studied differed in climatic and agrotechnical conditions during growth, which could have a significant impact on the amount of analyzed compounds. Contaminants from fertilizers used in conventional agriculture and the lower nutritional quality of the fruit could affect the low concentration of these compounds.

According to Lebedev, et al. [40], the anthocyanin content in 100 g of dry raspberry fruit *Polana* variety was 196.23 mg $100 \text{ g}^{-1}$ of cyanidin-3-glucoside. Similar results were obtained by other authors [42,44]. These results corresponded to conventionally grown raspberries from the 2020 crop. Other authors have obtained higher results for raspberry anthocyanin content. Stamenković, et al. [39] examining the species of the same raspberry variety obtained the results of 513.55 mg $100 \text{ g}^{-1}$ DW, which suggests that raspberries grown in Serbia may have a higher content of anthocyanins. The deep red color of the raspberry proper is related to its composition, including its anthocyanin content, the concentration of which is influenced by several factors, such as the variety, stage of ripening and climatic and soil characteristics of the growing sites, among others. Anthocyanins, and by extension raspberry color, are important to raspberry growers because they improve consumers' perception of quality. Sun exposure increases the concentration of these water-soluble pigments because it promotes the expression of flavonoid biosynthesis genes in the skin of these fruits. On the other hand, excessively high temperatures lead to inhibition of their biosynthesis [26]. In the results of this work, the highest anthocyanin content was obtained in organic raspberries. The reason for this may be that in organic production, plants activate their natural defense system against diseases and pests, so they synthesize more polyphenolic compounds, which have defensive functions in plants. Better health-promoting properties are also due to higher nutritional value and lower levels of contaminants, which are residues of agricultural chemicals from conventional farms [45]. The reason for the lower anthocyanin content of wild raspberry may have been, among other things, too little light [46].

Raspberry fruits tested by other researchers had lower antioxidant activity as determined by ferric-reducing antioxidant power [42]. Lebedev, et al. [40] showed that the *Polana* cv. of raspberry fruit had an antioxidant activity at the level of 156.60 mg TE $\text{g}^{-1}$ DW (ABTS assay) and 150.31 mg TE $\text{g}^{-1}$ DW (FRAP method). Our results obtained by the FRAP method are higher than those presented by the authors, while the results of antioxidant activity obtained by the ABTS assay were similar to the results of this authors.

Total carotenoids found in raspberries according to Belin, et al. [47] are 0.47 mg $\text{g}^{-1}$ DW. Polish nutritional tables [38] present these data at 0.02 mg/100 g (retinol + β-carotene).

Other studies have found that the $\alpha$-carotene content of raspberries ranged from 20 to 80, and beta carotene from 40 to 160 µg g$^{-1}$ DW; the seeds had a low carotene content, but were rich in zeaxanthin and lutein [48]. The carotenoid profile of raspberries is highly variable and depends on the species origin, fruit phenotype, growing site, sunlight and the period in which the fruit was harvested.

*3.4. Basic Composition of Raspberry Leaves*

Table 5 shows basic composition in the leaves of raspberries (*Rubus idaeus* L.) obtained from three crops: conventional, organic and wild grown.

**Table 5.** Basic composition in leaves of raspberries from conventional and organic practices as well as those wild grown (g 100 g$^{-1}$ DW).

| Basic Composition | Cultivation | Conventional | | Organic | | Wild Crop | |
|---|---|---|---|---|---|---|---|
| | Year | 2020 | 2021 | 2020 | 2021 | 2020 | 2021 |
| Water | | 66.06 ± 0.21 [b] | 66.05 ± 0.01 [b] | 67.63 ± 0.41 [a] | 68.37 ± 0.10 [a] | 68.49 ± 0.97 [a] | 60.16 ± 0.24 [c] |
| Total protein | | 18.18 ± 0.24 [a] | 12.85 ± 0.25 [b] | 18.08 ± 0.33 [a] | 14.98 ± 0.06 [d] | 17.90 ± 0.17 [a] | 14.28 ± 0.09 [c] |
| Crude fat | | 4.19 ± 0.16 [c] | 4.22 ± 0.18 [c] | 3.14 ± 0.11 [b] | 3.32 ± 0.03 [b] | 2.49 ± 0.18 [a] | 2.30 ± 0.12 [a] |
| Ash | | 9.05 ± 0.32 [e] | 7.36 ± 0.06 [c,d] | 7.05 ± 0.40 [b,c] | 6.59 ± 0.07 [a,b] | 7.87 ± 0.05 [d] | 6.07 ± 0.23 [a] |
| Dietary fibre | | 21.39 ± 0.34 [b] | 26.28 ± 0.02 [c] | 20.09 ± 0.33 [a] | 27.18 ± 0.34 [d] | 28.84 ± 0.41 [e] | 29.78 ± 0.26 [f] |
| Digestible carbohydrates | | 68.57 ± 0.08 [b] | 75.58 ± 0.12 [d] | 71.74 ± 0.04 [a] | 75.11 ± 0.02 [c] | 71.74 ± 0.30 [a] | 77.35 ± 0.26 [e] |

Results (mean ± SD) obtained from the analysis of three individual samples (n ≥ 3). Values within lines with different letters are significantly different (Duncan test $p \leq 0.05$).

The total protein levels in leaves of conventionally, organically and also wild raspberries picked in 2020 were substantially greater than in leaves from the 2021 season's harvest ($p \leq 0.05$). The total protein amount in the raspberry leaves collected in 2020 was similar, irrespective of their origin ($p > 0.05$). The protein content determined in the leaves of organic raspberries in 2021 was significantly higher than those of conventional and wild-growing leaves of raspberries. The statistically lowest protein content, in comparison to the other examined leaves, was found in the conventional raspberry leaves harvested in 2021. As for crude fat, its content in the conventional raspberry leaves was statistically higher than that determined in organic and wild-grown leaves of raspberries, regardless of the collect date. At the same time, its amount in the leaves of organic raspberries was significantly higher relative to the leaves of wild raspberries, regardless of the collection date. In turn, the highest statistically significant ash content ($p \leq 0.05$) was found in conventional raspberry leaves, and then in wild-growing raspberry leaves collected in 2020, compared to other samples. Ash content ($p \leq 0.05$) in the leaves of wild-growing raspberries from 2021 was the lowest and was statistically significant, compared to other examined leaves. Regardless of the year, dietary fibre content identified in leaves of field-grown raspberries was considerably larger ($p \leq 0.05$) compared to other leaves. In 2020, the dietary fibre amount in the organic raspberry leaves was statistically the lowest, compared to other investigated leaves. Interestingly, the highest statistically significant content of digestible carbohydrates was found in the wild-grown leaves of raspberry from 2021, while the lowest was found in those collected in 2020, compared to other leaves, in which their amount was similar ($p > 0.05$).

When comparing the values obtained by Biel and Jaroszewska [49], the raspberry leaves differed most in the fat content. The value given by the authors (7.05 g 100 g$^{-1}$ DW) was double our results. The protein content was comparable to the protein amount in the wild-grown leaves of raspberries and that from organic farming collected in 2021. According to Chwil and Kostryco [50], the protein content in the leaves of the raspberry cultivar *Glen Ample* was 21.43% and was higher compared to our findings. The fibre content determined by Biel and Jaroszewska [49] was almost two times lower (14.04 g 100 g$^{-1}$ DW),

as opposed to the results obtained by Chwil and Kostryco [50] who estimated its amount at 58%, which was twice as high. The fat and ash content in the raspberry leaves of the cultivar *Glen Ample*, reported by Chwil and Kostryco [50], was close to our results.

### 3.5. Selected Minerals Content in Raspberries Leaves

Selected minerals in the leaves of raspberry (*Rubus idaeus* L.) from various crops is presented (Table 6).

**Table 6.** Selected minerals in leaves of raspberries from conventional and organic practices as well as those wild grown (mg 100 g$^{-1}$ DW).

| Mineral Content | Cultivation | Conventional | | Organic | | Wild Crop | |
|---|---|---|---|---|---|---|---|
| | Year | 2020 | 2021 | 2020 | 2021 | 2020 | 2021 |
| Calcium (Ca) | | 1245.61 ± 32.09 [a] | 1967.18 ± 196.86 [b] | 1175.64 ± 24.48 [a] | 1097.29 ± 11.13 [a] | 1274.40 ± 90.13 [a] | 1255.23 ± 49.64 [a] |
| Iron (Fe) | | 14.27 ± 0.36 [a] | 14.67 ± 0.26 [a] | 12.23 ± 0.17 [c] | 10.39 ± 0.23 [b] | 14.04 ± 0.94 [a] | 23.61 ± 0.40 [d] |
| Potassium (K) | | 1171.58 ± 37.42 [d] | 1473.49 ± 95.11 [b] | 1381.33 ± 29.25 [a] | 2041.75 ± 9.50 [e] | 1435.38 ± 21.68 [a,b] | 1030.06 ± 33.98 [c] |
| Magnesium (Mg) | | 421.25 ± 20.81 [a] | 515.78 ± 43.59 [d] | 230.40 ± 4.67 [b] | 283.98 ± 4.37 [c] | 432.82 ± 21.16 [a] | 565.54 ± 28.34 [e] |
| Sodium (Na) | | 0.87 ± 0.02 [a] | 1.45 ± 0.27 [b,c] | 1.84 ± 0.43 [c] | 1.32 ± 0.08 [a,b] | 2.67 ± 0.42 [e] | 0.16 ± 0.07 [d] |
| Phosphorus (P) | | 356.67 ± 11.35 [a] | 699.84 ± 45.34 [e] | 160.62 ± 4.02 [b] | 343.87 ± 2.80 [a] | 291.90 ± 9.15 [d] | 203.87 ± 7.00 [c] |
| Sulphur (S) | | 124.04 ± 3.80 [d] | 152.52 ± 9.15 [b] | 114.45 ± 1.86 [c] | 174.83 ± 2.25 [e] | 143.61 ± 2.47 [a] | 144.75 ± 3.29 [a,b] |
| Selenium (Se) | | 0.0007 ± 0.000 [a,b] | 0.0036 ± 0.000 [d] | 0.0003 ± 0.000 [a] | 0.0011 ± 0.000 [b] | 0.0021 ± 0.000 [c] | 0.0046 ± 0.000 [e] |
| Barium (Ba) | | 1.69 ± 0.02 [b] | 1.93 ± 0.21 [b] | 1.27 ± 0.02 [a] | 1.08 ± 0.02 [a] | 4.40 ± 0.28 [c] | 4.89 ± 0.16 [d] |
| Lithium (Li) | | 0.0062 ± 0.001 [a] | 0.0131 ± 0.001 [c] | 0.0060 ± 0.001 [a] | 0.0048 ± 0.000 [a] | 0.0049 ± 0.002 [a] | 0.0106 ± 0.001 [b] |
| Beryl (Be) | | 0.0005 ± 0.000 [a] | 0.0006 ± 0.000 [a] | 0.0005 ± 0.000 [a] | 0.0001 ± 0.000 [c] | 0.0012 ± 0.000 [b] | 0.0015 ± 0.000 [b] |

Results (mean ± SD) obtained from the analysis of three individual samples (n ≥ 3). Values within lines with different letters are significantly different (Duncan test $p \leq 0.05$).

Calcium content was the highest and statistically significant ($p \leq 0.05$) in the conventional raspberry leaves harvested in 2021, compared to the remaining raspberry leaves, in which its content was similar ($p > 0.05$). As for iron, its content was the highest and significant in wild grown raspberry leaves from 2021 and the lowest in organic raspberry leaves from the 2021 harvest, compared to the other examined samples, which were generally similar ($p > 0.05$). An inverse relationship was observed in potassium content in 2021, as the highest significant amounts were in organic leaves and the lowest amounts in the leaves of wild-growing raspberries, compared to the other examined samples. The statistically highest magnesium content was noted in the leaves of wild grown raspberries and conventional raspberries in 2021, compared to the remaining leaves. Of all the leaves examined, the wild grown raspberry leaves contained the highest (2020) and lowest (2021) significant sodium concentrations. The highest statistically significant phosphorus and lithium contents, compared to the other samples, were in the conventional raspberry leaves from 2021. With regard to sulphur, its highest significant amount, compared to other samples, was in organic raspberry leaves from 2021 and the lowest also in organic raspberry leaves but collected a year earlier. The highest significant selenium amount, in comparison to the other samples, was in the leaves of wild-growing and conventional raspberries, collected in 2021. In turn, the lowest significant content ($p \leq 0.05$) of this micronutrient, compared to other examined leaves, was in the organic raspberry leaves from 2020. Beryllium content was the highest in the wild-grown raspberries leaves, regardless of the collect date; the lowest level was found in the leaves of organic raspberry from 2021, compared to other analysed samples, where the amounts of this component were similar.

The calcium amount in the examined leaves was greater compared to the literature data [49,51–53]. Researchers, who analysed different raspberry varieties, obtained similar and higher levels of this element, ranging between 1340 and 2110 mg 100 g$^{-1}$ DW [54]. Our results, apart from the calcium content in raspberry leaves from conventional culti-

vation in 2021, were within the ranges indicated by other authors [55]. The iron content obtained by different authors varied. The results they achieved were generally lower than ours [49,53,54]. However, our results were within the range specified by some authors [51]. The mean values (19.1 g 100 g$^{-1}$ DW) determined by Dresler, et al. [55] in the leaves of the cultivar *Polana* from 80 plantations in Lublin, are higher than our findings. Only the leaves of wild-growing raspberry harvested in 2021 had a higher iron content than the aforementioned results. Higher results were also obtained by Chwil and Kostryco [52] for the leaves of raspberry varieties *Glen Ample, Laszka* and *Radziejowa*, which were 17.6, 20.8 and 25.2 mg of iron per 100 g DW, respectively. The potassium content in the raspberry leaves indicated by Dresler [55] ranged from 1060 to 2050 g 100 g$^{-1}$ DW, which is consistent with our findings. Larger amounts of this element (the only exception was its content in the leaves from organic farming harvested in 2021) have been presented by other authors [49,51]. Lower and similar values of potassium content were obtained by [52–54]. In raspberry leaves from conventional cultivation and from wild-growing raspberries, magnesium content was higher than in those from organic farming. The results presented in the available literature correspond to the magnesium content of these two cultures [49,51,52,54,55]. However, in the studies of Karaklajić-Stajić [53] and Dresler [55], lower magnesium contents in the examined leaves were also found, amounting to 280 mg and 260 mg 100 g$^{-1}$ DW, respectively. As for sodium, the contents given by other authors are higher [49,51,54]. Of these, the lowest content, ranging in 26–41 mg 100 g$^{-1}$, was obtained by Horuz, et al. [54]. However, these results are still significantly higher than ours. On the other hand, the sodium contents in the leaves of three kinds of cultivars, determined by Chwil and Kostryco [52], were below 0.025 mg 100 g$^{-1}$ DW. The content of phosphorus in raspberry leaves reported in the literature is within the range of 140–400 mg 100 g$^{-1}$ DW. This is in line with the obtained data, except for the results noted in raspberry leaves from conventional cultivation in 2021, which contained 699.84 mg 100 g$^{-1}$ DW [49,51,53–55]. According to Chwil and Kostryco [52], leaves of three raspberry varieties contain between 209 and 255 mg phosphorus per 100 g of dry matter, which agree with our results referring to the leaves of wild-growing and organic raspberries from 2020. The sulphur content was determined by a single author and varied from 320 to 540 mg 100 g$^{-1}$ DW [54]. The results obtained were below this range.

Krzepiłko, et al. [56] examined the leaf buds of various varieties of *Rubus idaeus* L. raspberry—*Glen Fyne*, *Cascade Delight* and *Octavia*. That study showed that leaf buds of raspberries are characterized by a much lower content of calcium, potassium and magnesium, amounting to 188–219 mg, 609–638 mg and 89–91 mg 100 g$^{-1}$ DW, respectively. Only the iron content in the leaf buds was higher than that in raspberry leaves, and was between 16 and 23 mg 100 g$^{-1}$ DW.

### 3.6. Bioactive Compounds in Raspberries Leaves

The content of total polyphenols, carotenoids and anthocyanins as well as antioxidant activity of raspberry leaves from various kinds of cultivation different crops is shown in Table 7.

**Table 7.** Total polyphenol, carotenoids and anthocyanins content as well as antioxidant activity of the leaves of three raspberries cultivation at different harvest (2020 and 2021).

| Bioactive Compounds | Cultivation | Conventional | | Organic | | Wild crop | |
|---|---|---|---|---|---|---|---|
| | Year | 2020 | 2021 | 2020 | 2021 | 2020 | 2021 |
| Total polyphenols [mg CGA 100 g$^{-1}$ DW] | | 2480.45 ± 21.66 [b] | 4308.79 ± 36.80 [d] | 1873.02 ± 113.02 [a] | 3559.10 ± 100.62 [c] | 5078.39 ± 12.88 [e] | 8785.07 ± 303.74 [f] |
| Total polyphenols [mg GAE 100 g$^{-1}$ DW] | | 2838.75 ± 22.08 [b] | 5130.35 ± 39.76 [d] | 2223.23 ± 117.19 [a] | 4398.18 ± 110.37 [c] | 5554.47 ± 12.54 [e] | 9675.66 ± 314.94 [f] |
| ABTS$^{\bullet+}$ [μmol Trolox g$^{-1}$ DW] | | 67.21 ± 0.09 [a] | 802.49 ± 5.65 [b] | 70.86 ± 0.07 [a] | 852.32 ± 3.28 [c] | 72.79 ± 0.28 [a] | 988.63 ± 1.37 [e] |
| FRAP [μmol Trolox g$^{-1}$ DW] | | 492.49 ± 1.37 [b] | 987.78 ± 0.27 [d] | 418.74 ± 1.42 [a] | 702.93 ± 8.18 [c] | 1223.34 ± 40.01 [e] | 1967.93 ± 14.35 [f] |
| Total carotenoids [mg 100 g$^{-1}$ DW] | | 82.81 ± 4.38 [d] | 73.41 ± 2.95 [c] | 57.64 ± 3.56 [a,b] | 58.89 ± 2.56 [b] | 78.53 ± 3.89 [c,d] | 49.78 ± 2.31 [a] |

Results (mean ± SD) obtained from the analysis of three individual samples (n ≥ 3). Values within lines with different letters are significantly different (Duncan test $p \leq 0.05$).

Raspberry leaves, regardless of their cultivation type, were characterized by up to five times the amount of total polyphenols, with respect to raspberry fruit, regardless of source. The highest statistically significant content was characterized by leaves of wild raspberries, in which the amounts were up to four times higher, compared to the content of these compounds in, for example, leaves of conventional raspberries from 2020. The lowest content of the these compounds was found in the leaves of organic raspberry collected in 2020, relative to the other samples analyzed. Antioxidant activity, as measured by the ability to quench the cation radical ABTS, was highest in leaves of wild raspberries from 2021, compared to the other studied raspberry fruits. The lowest and similar antioxidant activity was found in raspberry leaves from the 2020 harvest, regardless of the origin of the leaves. These results generally coincide with those of the FRAP antioxidant activity determination. The only exception is the significantly higher antioxidant activity in wild raspberry leaves from 2020 versus to the results for leaves from other sources. The greatest content of total carotenoids was found in conventional raspberry leaves from 2020 and 2021, compared to the other examined raspberry leaves. The lowest level of these compounds was observed in wild raspberry leaves from 2021. Noteworthy is the almost 100 times higher content of these compounds in raspberry leaves in some cases, in comparison to fruit, regardless of origin.

Studies by other authors on plants of the *Rubus* family confirm that the content of phenolic compounds and thus their antioxidant activity in different parts of plants and their cultivars can vary due to the genotype of the plant and various cultivation factors, including environmental conditions [56]. In a study conducted on leaves of wild raspberry in the Balkans, a total polyphenol content of 59.68 to 96.83 mg per gallic acid in 1 g of fresh leaf weight was obtained [57]. Lebedev, et al. [40] examined the leaves of many raspberry varieties, including *Polana*, at various stages of growth: during flowering, fruit development and fruit ripening. The total polyphenol content of the *Polana* cultivar as calculated on GAE was 5890 mg 100 $g^{-1}$ DW. With the development of the fruit, the content of polyphenols in the leaves increased, from 6360 mg (flowering), to 3210 mg (fruit development) to the previously mentioned 5890 mg $100^{-1}$ g of DW for fruit ripening. Ponder and Hallmann [58], in their research, focused on determining the differences between conventional and organic cultivations of raspberry leaves. Conventional cultivation raspberries had a statistically lower total polyphenol content, which was not consistent with our results. The high content of total polyphenols in wild and organic raspberry leaves may be due to the fact that plants produce phenolic compounds to protect themselves against increased abiotic or biotic stresses, among other things. This is the plant's natural defense response against the harmful effects of stress factors, such as frost, drought, or intense light radiation, among others. In addition, these compounds protect plants from bacteria, viruses and fungi. Consequently, the plant can increase the production of polyphenols to protect against these organisms [58]. Most studies deal with the content of polyphenols in the raspberry fruit proper and their positive effects on the human body. However, comparing the content of total polyphenols in fruits and leaves of raspberries, it can be concluded that the latter have a significantly higher content of phenolic compounds and can be successfully used in phytotherapy.

Other studies also confirmed that the leaves are characterized by greater antioxidant activity than the fruits of the *Rubus idaeus* raspberry [59]. According to [58], leaves from organic farming had a higher content of antioxidant compounds determined by the $ABTS^{\bullet+}$ radical than those grown conventionally. Saki Toshima [42] in their work found that the determination of antioxidant activity using the ABTS method is significantly influenced by the amount of polyphenols, including flavonoids and ascorbic acid, while other polyphenolic compounds, e.g., anthocyanins contained in raspberry fruit, have a smaller effect. This is confirmed by our research, where the highest total polyphenol content was found in wild raspberry leaves, which also translated into the highest content of antioxidants (determined by the ABTS method). Other studies also showed that different harvest years have an influence on the content of compounds with antioxidant activity [59].

Studies have shown that the content of carotenoids in raspberry leaves from conventional cultivation is higher than in those from organic cultivation [58]. The results obtained in this study confirm this assumption. The high level of carotenoids in the leaves of conventional raspberries was due to the fact that conventional agriculture uses mineral fertilizers containing nitrogen, which in turn increases the production of compounds rich in this chemical element, such as, among other things, chlorophyll. In turn, the level of carotenoids in leaves depends primarily on their chlorophyll content. The higher the content of the green pigment in the leaves, the higher the carotenoid content is found in the plant. This is due to the fact that carotenoids have a protective function over the photosynthetic system against photo-oxidation during photosynthesis [58].

Anthocyanins were not found in raspberry leaves, which was confirmed by research earlier [40,60]. Other studies also reported that the leaves of the berries did not contain them [40]. Moreover, the literature shows that the type of solvent used and the extraction conditions may determine the attendance of anthocyanins in the leaves of berry plants [61].

There are no literature data, as far as we now know, on the content of selenium, barium, lithium, and beryllium in the fruits and leaves of organically or conventionally grown raspberries, or in those from the wild.

The basic composition of the raspberry fruits and leaves differs from one another. Raspberry fruits have a lower content of protein and ash, and higher level of dietary fibre and carbohydrates. The biggest difference is the amount of protein, whose content in leaves is two to three times higher. The difference in fat content is not very large, however, raspberry fruits may contain more due to the presence of seeds. Raspberry leaves have been found to have a higher mineral content than raspberry fruits.

The results presented may vary, as raspberries were exposed to different climatic conditions, such as weather and soil. The chemical composition also depends on the variety, the stage of maturity at harvest, and the storage conditions [62]. In addition, there are parameters such as different insolation, the location of the fruit on the shrub, and the harvest date, which may also affect the results [63]. Temperature and rainfall significantly influence the chemical constitution of the fruit; however, the accumulation of selected components also depends strongly on the genotype [63–65]. Since the quality of the fruit is determined during the growth and maturation of the fruit, factors such as light intensity, temperature, rainfall, wind, and pests play a significant role [66].

One of the factors causing the differences in the content of elements is the soil pH, since the strongly acidic reaction contributes to the significant uptake of elements from the soil by the fruit [33]. In addition, it has been shown that alkaline soils, predominating in temperate regions, including Europe, have greater levels of P, Ca, K and Mg and also a lower iron content than acidic soils [27]. However, Hakala, et al. [67], who examined six strawberry cultivars for two years, proved that variety had a bigger impact on the mineral content than climate and soil conditions alone. Stojanov, et al. [68] surveyed the connection between the use of mineral, organic, and organic-mineral fertilisers on the quality characteristics of the *Rubus idaeus* L. raspberry fruit. Some fertilisers improved yields, as well as the content of bioactive substances, and therein the total content of flavonoids and phenols, as well as antioxidant activity. This may indicate that different growing conditions, including the use of various fertilisers, can affect nutritional and non-nutritional content in the fruits and leaves of plants. Anjos, et al. [69] studied the consequences of conventionally and organically oriented agricultural practices on the plant-chemistry content of raspberries. According to the authors, this relationship is obvious, but it depends on the variety.

The accumulation of macro- and micronutrients depends on the cultivation method used and the level of nutrients in the soil. The cultivation of raspberries on a soilless nutrient medium increased the content of potassium, sodium and, selenium by one and a half to three times compared to conventional cultivation. Conventional cultivation, however, resulted in a higher accumulation of calcium and iron by 2 and 7.4 times, respectively [70].

Moreover, differences in the nutritional quality of the fruits may be due to various cultivation methods that affect the taxonomic and functional composition of the bacterial

microbiome. Bacterial biodiversity in organic farming was shown to be lower, resulting in fruits with a higher content of health-promoting compounds (anthocyanins) [71]. Thus, the question is whether the bacterial microbiome also affects the content of other nutrients and non-nutrients in plant raw materials.

## 4. Conclusions

The quantities of total protein and crude fat were generally higher in organic and wild raspberry fruits harvested in 2020 and 2021, in comparison to those from conventional farming practices. The highest significant amounts of dietary fibre were found in wild grown raspberries collected both in 2020 and 2021, compared to the other raspberries tested. Digestible carbohydrates were determined in the highest, statistically significant amounts in conventional raspberries from 2020 and 2021, as well as wild raspberries from 2021, compared to the other fruits, for which the results were similar. The basic composition and mineral content of raspberry fruits and leaves are fundamentally different. The total protein levels in leaves of conventionally, organically, and wild raspberries picked in 2020 were substantially greater than in leaves from the 2021 season's harvest. Raspberry fruits had less protein and ash, and more dietary fibre and carbohydrates, in comparison to raspberry leaves. The biggest difference was the amount of protein, which was two to three times larger in the leaves. The mineral content of raspberry leaves was found to be greater than that of raspberry fruits. Raspberry leaves, regardless of their cultivation type, were characterized in some cases by up to five times the amount of total polyphenols and 100 times higher of total carotenoids, with respect to raspberry fruit, regardless of source. Antioxidant activity, measured by ABTS and FRAP methods, was greatest in raspberry fruits and leaves from 2021 versus the other examined samples from 2022.

It has not been unequivocally established which cultivation method is the most favorable in terms of basic nutrient levels, selected mineral content, or antioxidant properties. Further research in this area is needed.

**Author Contributions:** Conceptualization, M.K. and J.K.-D.; methodology, M.K., J.K.-D., S.S.; investigation, M.K., J.K.-D., S.S. and I.D.; writing—original draft preparation, M.K., J.K.-D.; writing—review and editing, J.K.-D.; supervision, J.K.-D.; funding acquisition, M.K. All authors have read and agreed to the published version of the manuscript.

**Funding:** This research received no external funding.

**Institutional Review Board Statement:** Not applicable.

**Informed Consent Statement:** Not applicable.

**Data Availability Statement:** Not applicable.

**Conflicts of Interest:** The authors declare no conflict of interest.

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
