# Peer review of "Phytochemical Composition of the Fruits and Leaves of Raspberries (Rubus idaeus L.)—Conventional vs. Organic and Those Wild Grown"

_applsci, doi:10.3390/app122211783_

Round 1

Reviewer 1 Report

Raspberry (Rubus Idaeus) is a forest fruit of a shrub belonging to the Rosaceae family. It has numerous beneficial effects, in fact it is a natural anti-inflammatory and purifying agent, it also has a laxative action. It also improves circulation and protects capillaries. In this study, raspberry fruits and leaves from organic, conventional and wild crops were analyzed for total protein content of dry matter, crude fat, ash, dietary fiber, digestible carbohydrates and selected important essential minerals, as well as several trace elements.

The topic of the manuscript is not particularly original, in fact several articles of literature report the characterization of the raspberry. However, in this work the authors compared the composition of raspberry fruits and leaves obtained from different types of crops. Furthermore, the experimental part of the work focused on the determination of numerous heavy metals present in these matrices.

The manuscript is well organized. The experimental part provides new data for the scientific literature on the topic.

The references are up-to-date and in sufficient numbers.

However, some changes are required, as follows:

Check all the text: some recurring terms must always be written in the same way, for example dietary fiber or dietary fiber (line 17).

Line 21: write in full the names of the elements (bar and lit); (check and standardize throughout the text, including tables).

The conclusions section should be enriched with the results obtained.

Author Response

The authors are very grateful to the anonymous Referees for their valuable comments.

Answers for detailed comments of Reviewers

Open Review

( ) I would not like to sign my review report
(x) I would like to sign my review report English language and style

( ) Extensive editing of English language and style required
( ) Moderate English changes required
(x) English language and style are fine/minor spell check required
( ) I don't feel qualified to judge about the English language and style

Yes

Can be improved

Must be improved

Not applicable

Does the introduction provide sufficient background and include all relevant references?

(x)

( )

( )

( )

Are all the cited references relevant to the research?

(x)

( )

( )

( )

Is the research design appropriate?

(x)

( )

( )

( )

Are the methods adequately described?

(x)

( )

( )

( )

Are the results clearly presented?

( )

(x)

( )

( )

Are the conclusions supported by the results?

( )

(x)

( )

( )

Comments and Suggestions for Authors

Raspberry (Rubus Idaeus) is a forest fruit of a shrub belonging to the Rosaceae family. It has numerous beneficial effects, in fact it is a natural anti-inflammatory and purifying agent, it also has a laxative action. It also improves circulation and protects capillaries. In this study, raspberry fruits and leaves from organic, conventional and wild crops were analyzed for total protein content of dry matter, crude fat, ash, dietary fiber, digestible carbohydrates and selected important essential minerals, as well as several trace elements.

The topic of the manuscript is not particularly original, in fact several articles of literature report the characterization of the raspberry. However, in this work the authors compared the composition of raspberry fruits and leaves obtained from different types of crops. Furthermore, the experimental part of the work focused on the determination of numerous heavy metals present in these matrices.

The manuscript is well organized. The experimental part provides new data for the scientific literature on the topic.

The references are up-to-date and in sufficient numbers.

However, some changes are required, as follows:

Check all the text: some recurring terms must always be written in the same way, for example dietary fiber or dietary fiber (line 17). Unified

Line 21: write in full the names of the elements (bar and lit); (check and standardize throughout the text, including tables). Corrected.

The conclusions section should be enriched with the results obtained. Done

Reviewer 2 Report

The authors investigated comparatively the contents of proteins, ash, fat, carbohydrate, fibre and minerals in leaf and fruit samples obtained from raspberries cultivated conventionally and organically, as well as those growing wild. Based on the data collected, no conclusion was drawn on which cultivation was the most favorable. The way of writing is convoluted and unconcise, often making it not easy to follow or incoherent. In the data presented, some standard deviations are extremely high (sometimes even greater than the mean values). These must be double-checked to see whether it was due to typo errors or due to large variations in samples collected, which would then necessitate the inclusion of more than three replicates in the analyses.

Below are my feedbacks for the authors’ consideration:

1.     ABSTRACT

·       This should be revised to make it more informative, quantitative, and specific.

·       It would be useful to add a concluding remark too.

·       In lines 27-30, it is unclear whether the descriptions refer to organic, conventional, or wild raspberries.

·       Also, instead of saying “… lower content of protein and ash, and higher level of dietary fibre and carbohydrates” and “… a higher mineral content than …”, the authors can consider revising them to add in more quantitative information.

2.     M&M:

·       Please indicate more specific details regarding the time/date of sample collection in line 84 (“The material came from a 2-year collection”).

·       Not knowing the variety of the wild raspberries could weaken the comparison the authors tried to make. This should be clearly spelt out in the discussion.

·       Is it possible to indicate the GPS locations of the site of sample collection?

3.     RESULTS & DISCUSSION

·       The first paragraph mentioned water content, but no data were shown. So, it is not possible to check whether the water content data are what the authors described.

·       Tables 1 and 2 – In both, the unit for “digestible carbohydrates” (5th row) is missing. Please check.

·       Tables 1 and 2 – In both, the footnotes say “n=3” (lines 152 and 269). But in M&M, line 137 (“at least three replications”) suggests that in at least some of their analyses, the authors used more than three replicates. Please check which one is correct - the footnotes or the M&M.

·       In Table 1, the superscript letters used for indicating statistical significance should be placed after the SD, not after the mean values. Also, “ab” should be “a,b” instead – a comma is needed.

·       In Table 1, please check data where the SD values are greater than (sometimes > 10 times larger than!) the mean values, i.e., most of mineral data for the conventional-2020 group, except for Fe, Se, and Ba. There are other data with such issues in the other groups too. This seems unusual. Please check whether it was due to typo errors or due to large variations in the samples collected. If it is the latter, the authors should consider using more than three replicates in the analyses.

·       In Table 1, inconsistencies in using decimals should be checked too. For example, for Ba, mean values are presented with sometimes two, sometimes three decimal places. The same applies to the SD values. This makes the choice of decimal places used seems random.

·       In Table 1, for Li and Be, the SD values are presented with fewer decimal places than the mean values. Please check.

·       For Table 1, the footnote mentions “one-way ANOVA” (line 154), but the M&M mentions only “Two-way analysis of variance (ANOVA)” in line 138. Please check.

·       Please also check Table 2 for the aforementioned issues.

·       Lines 156-165 – Please rewrite to make it more concise and clearer.

·       Lines 165-168 – The statement seems inaccurate, based on the statistical info in the table, the conventional-2020 group is similarly the highest as the two groups mentioned by the authors (“conventional raspberries from 2020 and wild-growing ones from 2021”). Please recheck.

·       Lines 237-243 – This paragraph seems disconnected from the rest of the discussion on the authors’ data. It simply looks like literature review. Please try to discuss data obtained in relation to info in this paragraph. Also, in the paragraph, it is unclear what comparisons the “lower”/ “higher” descriptions (lines 238, 239, 241) refer to.

·       In general, although the authors made an effort to discuss their findings with those of others, the comparison seems not in-depth/convincing enough. For example, in lines 169-175, it is unclear whether the authors actually compared their data on the organic group to others’ data which are also on organically cultivated plants.

Author Response

The authors are very grateful to the anonymous Referees for their valuable comments.

Open Review

English language and style

( ) Extensive editing of English language and style required
(x) Moderate English changes required
( ) English language and style are fine/minor spell check required
( ) I don't feel qualified to judge about the English language and style

Yes

Can be improved

Must be improved

Not applicable

Does the introduction provide sufficient background and include all relevant references?

( )

(x)

( )

( )

Are all the cited references relevant to the research?

( )

(x)

( )

( )

Is the research design appropriate?

( )

(x)

( )

( )

Are the methods adequately described?

( )

(x)

( )

( )

Are the results clearly presented?

( )

( )

(x)

( )

Are the conclusions supported by the results?

( )

( )

(x)

( )

Comments and Suggestions for Authors

The authors investigated comparatively the contents of proteins, ash, fat, carbohydrate, fibre and minerals in leaf and fruit samples obtained from raspberries cultivated conventionally and organically, as well as those growing wild. Based on the data collected, no conclusion was drawn on which cultivation was the most favorable. The way of writing is convoluted and unconcise, often making it not easy to follow or incoherent. In the data presented, some standard deviations are extremely high (sometimes even greater than the mean values). These must be double-checked to see whether it was due to typo errors or due to large variations in samples collected, which would then necessitate the inclusion of more than three replicates in the analyses.

Below are my feedbacks for the authors’ consideration:

  1. ABSTRACT
  • This should be revised to make it more informative, quantitative, and specific. Corrected
  • It would be useful to add a concluding remark too. Added
  • In lines 27-30, it is unclear whether the descriptions refer to organic, conventional, or wild raspberries. Corrected
  • Also, instead of saying “… lower content of protein and ash, and higher level of dietary fibre and carbohydrates” and “… a higher mineral content than …”, the authors can consider revising them to add in more quantitative information. Due to the large number of results, it is not possible to add quantitative value to the abstract, which is limited to 200 words.

  1. M&M:
  • Please indicate more specific details regarding the time/date of sample collection in line 84 (“The material came from a 2-year collection”). Corrected
  • Not knowing the variety of the wild raspberries could weaken the comparison the authors tried to make. This should be clearly spelt out in the discussion. Corrected in the section of discussion

Authors comment: There are several hundred types of raspberry known in the world and their number is constantly growing. Several dozen are cultivated on a larger scale. In Poland, only seedlings of the varieties entered in the register may be sold, because it is an economically important species. Nursery material is produced under the control of the State Seed Inspection and the State Plant Protection Service. The wild grown are not under control. We don’t know exactly the varieties of wild growing Rubus idaeus species. The research was intended to be consumer-based, and therefore the variety of wild raspberries is not significantly influential. The authors wanted to broaden the scope of the study as to whether potential consumers should reach for wild raspberry fruit and whether it could provide an alternative source of raspberry choice to conventional or organic raspberries. Edible fruit species containing a great amount of various vitamins, antioxidants and minerals are of a significant importance in creating biodiversity of species.

  • Is it possible to indicate the GPS locations of the site of sample collection? Added.

  1. RESULTS & DISCUSSION
  • The first paragraph mentioned water content, but no data were shown. So, it is not possible to check whether the water content data are what the authors described. The water content of the fruit was added to the first row of the table.
  • Tables 1 and 2 – In both, the unit for “digestible carbohydrates” (5th row) is missing. Please check. Corrected
  • Tables 1 and 2 – In both, the footnotes say “n=3” (lines 152 and 269). But in M&M, line 137 (“at least three replications”) suggests that in at least some of their analyses, the authors used more than three replicates. Please check which one is correct - the footnotes or the M&M. Corrected.
  • In Table 1, the superscript letters used for indicating statistical significance should be placed after the SD, not after the mean values. Also, “ab” should be “a,b” instead – a comma is needed. Corrected
  • In Table 1, please check data where the SD values are greater than (sometimes > 10 times larger than!) the mean values, i.e., most of mineral data for the conventional-2020 group, except for Fe, Se, and Ba. There are other data with such issues in the other groups too. This seems unusual. Please check whether it was due to typo errors or due to large variations in the samples collected. If it is the latter, the authors should consider using more than three replicates in the analyses. Corrected. It resulted from numerical errors and an incorrectly shifted comma.
  • In Table 1, inconsistencies in using decimals should be checked too. For example, for Ba, mean values are presented with sometimes two, sometimes three decimal places. The same applies to the SD values. This makes the choice of decimal places used seems random. Corrected. All values (except selenium, lit and beryl) have two decimal places. Due to the low values of selenium, lit and beryl have values with 4 decimal places and SD with 3 decimal places.
  • In Table 1, for Li and Be, the SD values are presented with fewer decimal places than the mean values. Please check. Checked and corrected
  • For Table 1, the footnote mentions “one-way ANOVA” (line 154), but the M&M mentions only “Two-way analysis of variance (ANOVA)” in line 138. Please check. Corrected. Two-way ANOVA was performed everywhere.
  • Please also check Table 2 for the aforementioned issues. Corrected in the same way as above.
  • Lines 156-165 – Please rewrite to make it more concise and clearer. Corrected
  • Lines 165-168 – The statement seems inaccurate, based on the statistical info in the table, the conventional-2020 group is similarly the highest as the two groups mentioned by the authors (“conventional raspberries from 2020 and wild-growing ones from 2021”). Please recheck. Checked.
  • Lines 237-243 – This paragraph seems disconnected from the rest of the discussion on the authors’ data. It simply looks like literature review. Please try to discuss data obtained in relation to info in this paragraph. Also, in the paragraph, it is unclear what comparisons the “lower”/ “higher” descriptions (lines 238, 239, 241) refer to. Corrected.

·       In general, although the authors made an effort to discuss their findings with those of others, the comparison seems not in-depth/convincing enough. For example, in lines 169-175, it is unclear whether the authors actually compared their data on the organic group to others’ data which are also on organically cultivated plants. Checked and improved

Reviewer 3 Report

The MS deals only with proximate analysis and mineral analysis, the following test may improve the novelty aspect of the MS: 1. Phenolic profiling 2. Antioxidant characterisation. 3. Microstructural aspects (SEM) 4. In-vitro medicinal properties 5. In-vitro gastrointestinal digestion characteristics.

Author Response

The authors are very grateful to the anonymous Referees for their valuable comments.

Open Review

English language and style

( ) Extensive editing of English language and style required
(x) Moderate English changes required
( ) English language and style are fine/minor spell check required
( ) I don't feel qualified to judge about the English language and style

Yes

Can be improved

Must be improved

Not applicable

Does the introduction provide sufficient background and include all relevant references?

( )

(x)

( )

( )

Are all the cited references relevant to the research?

( )

( )

(x)

( )

Is the research design appropriate?

( )

( )

(x)

( )

Are the methods adequately described?

( )

(x)

( )

( )

Are the results clearly presented?

( )

( )

(x)

( )

Are the conclusions supported by the results?

( )

( )

(x)

( )

Comments and Suggestions for Authors

The MS deals only with proximate analysis and mineral analysis, the following test may improve the novelty aspect of the MS: 1. Phenolic profiling 2. Antioxidant characterisation. 3. Microstructural aspects (SEM) 4. In-vitro medicinal properties 5. In-vitro gastrointestinal digestion characteristics.

Dear Reviewer,

You are absolutely right that I could additionally make more research and can improve the knowledge in this area.

Therefore, I would like to assure You that further publications are planned on the above subject, which will include:

  • toxicity testing with the CaCO2, HT-29, HEP G2 and CCD 841 CoN cell lines, adehsion ability with probiotics of the research material on the Caco2 cell line using rhamnosus, L. plantarum, L. gassery, L. reuteri,
  • phenolic profiling using HPLC-DAD-MS and HPTLC,
  • the content of total carotenoids [PN-90/A-75101/12], the content of anthocyanins [Benvenuti et al. 2004], the content of total polyphenols [Swain and Hillis, 1959, with its own modifications], antioxidant activities as based on the mechanism of ABTS•+ free radical scavenging [Re et al., 1999] and determination of antioxidant activity by FRAP method [Benzie and Strain 1996].

All studies will refer to organic, conventional and wild raspberry leaves and fruits for 2-3 years of cultivation.

Due to the requirements of the publishing house and the large amount of data that could be illegible in one publication, we decided to divide the above results into 2-3 publications.

Round 2

Reviewer 3 Report

Previous comments not adddressed

Author Response

Dear Reviewer,

authors improved the text of manuscript accordance with the notes of Reviewer - we have improved the publication according to Your suggestions.

In addition, we have enriched the work with the following analyses: the content of total carotenoids, the content of anthocyanins, the content of total polyphenols, antioxidant activities as based on the mechanism of ABTS•+ free radical scavenging and determination of antioxidant activity by FRAP method.

The authors are very grateful to the anonymous Reviewer for Your valuable comments.

Yours sincerely,

Joanna Kapusta-Duch